MULTI-DEEP: A novel CAD system for coronavirus (COVID-19) diagnosis from CT images using multiple convolution neural networks

Attallah Omneya o.attallah@aast.edu
http://orcid.org/0000-0002-4455-2507 Ragab Dina A. dinaragab@aast.edu
Sharkas Maha
Electronics and Communications Engineering Department, Arab Academy for Science, Technology, and Maritime Transport (AASTMT) , Alexandria , Egypt
Aly Sharif
Electronic publication date: 2020 Sep 30
Publication date: 2020
Volume: 8
Electronic Location ID: e10086
Received 2020 May 15; Accepted 2020 Sep 11
Copyright: © 2020 Attallah et al.
Copyright year: 2020
Copyright holder: Attallah et al.
License: This is an open access article distributed under the terms of the Creative Commons Attribution License, which permits unrestricted use, distribution, reproduction and adaptation in any medium and for any purpose provided that it is properly attributed. For attribution, the original author(s), title, publication source (PeerJ) and either DOI or URL of the article must be cited.
License URL: https://creativecommons.org/licenses/by/4.0/

Keywords: Computed tomography (CT), Computer-aided diagnosis (CAD), Convolution neural networks (CNN), Coronavirus (COVID-19), Deep learning (DL).

Funding: The authors received no funding for this work.

==============================
Coronavirus (COVID-19) was first observed in Wuhan, China, and quickly propagated worldwide. It is considered the supreme crisis of the present era and one of the most crucial hazards threatening worldwide health. Therefore, the early detection of COVID-19 is essential. The common way to detect COVID-19 is the reverse transcription-polymerase chain reaction (RT-PCR) test, although it has several drawbacks. Computed tomography (CT) scans can enable the early detection of suspected patients, however, the overlap between patterns of COVID-19 and other types of pneumonia makes it difficult for radiologists to diagnose COVID-19 accurately. On the other hand, deep learning (DL) techniques and especially the convolutional neural network (CNN) can classify COVID-19 and non-COVID-19 cases. In addition, DL techniques that use CT images can deliver an accurate diagnosis faster than the RT-PCR test, which consequently saves time for disease control and provides an efficient computer-aided diagnosis (CAD) system. The shortage of publicly available datasets of CT images, makes the CAD system’s design a challenging task. The CAD systems in the literature are based on either individual CNN or two-fused CNNs; one used for segmentation and the other for classification and diagnosis. In this article, a novel CAD system is proposed for diagnosing COVID-19 based on the fusion of multiple CNNs. First, an end-to-end classification is performed. Afterward, the deep features are extracted from each network individually and classified using a support vector machine (SVM) classifier. Next, principal component analysis is applied to each deep feature set, extracted from each network. Such feature sets are then used to train an SVM classifier individually. Afterward, a selected number of principal components from each deep feature set are fused and compared with the fusion of the deep features extracted from each CNN. The results show that the proposed system is effective and capable of detecting COVID-19 and distinguishing it from non-COVID-19 cases with an accuracy of 94.7%, AUC of 0.98 (98%), sensitivity 95.6%, and specificity of 93.7%. Moreover, the results show that the system is efficient, as fusing a selected number of principal components has reduced the computational cost of the final model by almost 32%.

Introduction

Late December 2019, the coronavirus disease (COVID-19) was detected in Wuhan City, Hubei Province, China. It then propagated worldwide, and was declared a pandemic by the World Health Organization (WHO) at the end of January 2020 (Butt et al., 2020). The pandemic of the novel coronavirus is still progressively increasing and propagating around the world. So far, the pandemic has spread to more than 30 provinces of China and other countries (Li et al., 2020b). The number of patients that were verified to have COVID-19 has increased to over 775,000 in more than 160 countries, but the number of persons diseased is likely to be much greater. Over 36,000 individuals have died from the COVID-19 disease (up to 30 March 2020) (Dong, Du & Gardner, 2020).

Nowadays, COVID-19 is considered the most significant and crucial hazard to worldwide health (Dong, Du & Gardner, 2020). The WHO encouraged efforts to reduce the spread of the infection, however, many countries have faced some major care crises (Arabi, Murthy & Webb, 2020; Grasselli, Pesenti & Cecconi, 2020). The epidemic of COVID-19 led to a tremendous increase in the need for hospital beds, as well as necessary medical equipment. Moreover, doctors and nurses are at a high risk of infection. The new coronavirus has led to severe health problems such as heart complications, acute respiratory condition, and secondary infections, in quite a great portion of infected people. It consequently led to significant deaths. Therefore, the early detection and initiation of treatment in severe cases is essential (Butt et al., 2020).

The reverse transcription-polymerase chain reaction (RT-PCR) test is a widely used tool to diagnose COVID-19 cases. However, it suffers from some drawbacks, such as its time-consuming procedure and expensive cost. Moreover, the sensitivity of the RT-PCR test is inadequate, resulting in false negatives, and subsequently more infections (Ai et al., 2020). Furthermore, the insufficient amount of tests and the need for well-equipped laboratories for analysis, could critically delay the accurate detection of COVID-19 cases. This shows the unexpected challenges faced, in trying to avert the spread of the virus globally (Xie et al., 2020). These limitations have encouraged researches to look for a faster alternative solution to detect coronavirus.

Deep learning (DL) techniques are considered the new class of machine learning (ML) approaches. The explanation of different DL techniques will be discussed later in the Materials and Methods section. Recently, DL has been extensively used in the medical area (Zemouri, Zerhouni & Racoceanu, 2019). This is due to the advantage they offer over classical ML approaches. DL methods have the ability to present optimum representations and significant information from the raw images without image enhancement, segmentation, and feature extraction processes. This leads to an improved diagnosis process and lower complexity, when compared to classical ML approaches (Attallah, Sharkas & Gadelkarim, 2020; Fujita, 2020). Consequently, DL based-CAD systems that use medical images are recommended as another tool in the diagnosis and control of the novel coronavirus. Such CAD systems offer a quick and easy solution for medical diagnosis of COVID-19 (Iwasawa et al., 2020). Moreover, these CAD systems have the potential to deliver an accurate and fast second opinion, to help radiologists in giving an accurate diagnosis based on medical images (Ardakani et al., 2020). Furthermore, a CAD system can avoid diagnostic errors caused by human which might occur due to the exertion done during clinical examinations and radiologists’ visual fatigue (Ragab, Sharkas & Attallah, 2019; Ragab et al., 2019). This can assist in managing the current pandemic, speed up the detection of the disease, avoid its fast spread, and help doctors to improve the quality of patient management, even in extraordinary workload situations.

Computed tomography (CT) images are well-known imaging technique that can be employed by CAD systems for the early detection of the COVID-19 disease, and distinguishing it from non-COVID-19 cases (Butt et al., 2020). The authors in Fang et al. (2020) have shown that the sensitivity of detecting COVID-19 using CT medical images is much higher than the RT-PCR test. Several studies have demonstrated that CT is an efficient tool in detecting, visualizing, and diagnosing COVID 19-disease (Fang et al., 2020; Xie et al., 2020). The representative appearance of COVID-19 in CT images allows such images to differentiate COVID-19 from non-COVID-19 pneumonia. This task is difficult when it depends only on human examination. This is because of the overlap of specific patterns with other types of pneumonia (Shi, Han & Zheng, 2020). However, DL techniques can distinguish between COVID-19 and non-COVID-19 cases (Butt et al., 2020). Therefore, DL methods could be used to enhance the radiologist’s diagnostic capabilities for detecting COVID-19 from CT images.

The authors in Butt et al. (2020) proposed the use of ResNet-23 and ResNet-18 convolutional neural network (CNN) architectures. They differentiated COVID-19 and non-COVID-19 cases achieving an area under the receiver-operating curve (AUC) of 99.6%, a sensitivity of 98.2%, and a specificity of 92.2%. The authors in Li et al. (2020a) introduced a method based on ResNet-50 to detect COVID-19 from CT images. The sensitivity, specificity, and AUC produced were 87%, 92%, and 0.95 (95%), respectively. EfficientNet-B4 was used in Bai et al. (2020) to distinguish between corona and non-coronavirus achieving 87%, 89%, 86%, and 0.9 (90%) for accuracy, sensitivity, specificity, and AUC, respectively. On the other hand, the authors in Amyar, Modzelewski & Ruan (2020) constructed two U-Nets. The highest accuracy, sensitivity, specificity, and AUC attained were 86%, 94%, 79%, and 0.93 (93%), respectively. The authors in Chen et al. (2020) employed the U-Net architecture to distinguish between COVID-19 and non-COVID-19 cases. A sensitivity of 100%, a specificity of 93.6%, and an accuracy of 95.2% were obtained. A U-Net was used for lung segmentation in Zheng et al. (2020) followed by a 3D CNN for predicting the probability of COVID-19. The proposed method reached a sensitivity of 90.7%, a specificity of 91.1%, and an AUC of 0.959 (95.9%). Whereas, a 2D Deeplab-v1 model was used in Jin et al. (2020a) for segmenting the lung followed by the ResNet-152 model for identifying COVID-19 cases. The results achieved a sensitivity of 94.1%, a specificity of 95.5%, and an AUC of 0.979 (97.9%). The authors in Jin et al. (2020b) proposed the fusion of UNet++ for segmenting lesions and a ResNet-50 for classification. The sensitivity and specificity of ResNet-50 CNN were 97.4% and 92.2%, respectively. Moreover, ResNet-50 CNN was also employed in Song et al. (2020), achieving an accuracy of 86%, a sensitivity of 96%, a precision of 79%, and an AUC of 0.95 (95%). The authors in Wang et al. (2020) extracted the deep features using Inception CNN architecture and then used these features to train Adaboosted decision tree classifiers. An accuracy of 82.9%, a sensitivity of 81%, a specificity of 84%, and an AUC of 0.9 (90%) were attained.

A summary of recent similar related studies is shown in Table 1. As it is clear some of the related work is based on either an individual CNN or fusing two CNNs, where the first CNN is for segmenting the lung and the other is for the classification and diagnosis of COVID-19. The effect of fusing multiple CNNs for classification and diagnosis was not considered in the previous studies. Moreover, reducing the huge deep feature space was not examined in the previous studies. Therefore, in this article, a novel CAD system based on the fusion of multiple CNN for detecting COVID-19 and differentiating it from non-COVID-19 cases is proposed. The proposed CAD employs four types of CNNs different from those used in the literature. These CNNs include AlexNet, GoogleNet, ResNet-18, and Shuffle-Net. The proposed CAD examines the use of principal component analysis (PCA), to reduce the dimensionality of the deep feature space extracted from each CNN individually, and use them to train support a vector machine (SVM) classifier. Then, the effect of fusing these principal components on the classification performance of the SVM classifier is evaluated. The proposed CAD system classifies COVID-19 and non-COVID-19 by four different scenarios. The first one is an end-to-end CNN using four fine-tuned pre-trained CNNs. In the second scenario, the deep features of each CNN architecture is extracted and classified separately using the SVM classifier. In the third scenario, selected principal components are chosen from each deep feature set, and used to train SVM. Finally, in the fourth scenario, all principal components selected from the four deep features sets, are fused and classified using SVM.

Table 1 A summary of recent related studies.

Paper	Dataset	Method	Results	
Butt et al. (2020)	219 COVID-19
339 Others	ResNet-18 and ResNet-23	Accuracy = 95.2%	
AUC = 0.996 (99.6%)	
Sensitivity = 98.2%	
Specificity = 92.2%	
Li et al. (2020a)	468 COVID-19
2,996 Others	ResNet-50	Accuracy = 89.5%	
AUC = 0.95 (95%)	
Sensitivity = 87%	
Specificity = 92%	
Bai et al. (2020)	521 COVID-19
665 others	EfficientNet-B4	Accuracy = 87%	
AUC = 0.9 (90%)	
Sensitivity = 89%	
Specificity = 86%	
Amyar, Modzelewski & Ruan (2020)	449 COVID-19
595 others	Two U-Nets	Accuracy = 86%	
AUC = 0.93 (93%)	
Sensitivity = 94%	
Specificity = 79%	
Chen et al. (2020)	4382 COVID-19
9,369 others	U-Net++	Accuracy = 95.2%	
Sensitivity = 100%	
Specificity = 93.6	
Zheng et al. (2020)	313 COVID-19
229 Others	U-Net and CNN	Accuracy = 90.9%	
AUC = 0.959 (95.9%)	
Sensitivity = 90.7%	
Specificity = 91.1%	
Jin et al. (2020a)	496 COVID-19
1,385 Others	Deeplab-v1 and ResNet-152	Accuracy = 94.8%	
AUC = 0.979 (97.9%)	
Sensitivity = 94.1%	
Specificity = 95.5%	
Jin et al. (2020b)	723 COVID-19
413 Others	U-Net and ResNet-50	Accuracy = 94.8%	
Sensitivity = 97.4%	
Specificity = 92.2%	
Song et al. (2020)	219 COVID-19
399 Others	ResNet-50	Accuracy = 86%	
AUC = 0.95 (95%)	
Sensitivity = 96%	
Precision = 79%	
Wang et al. (2020)	325 COVID-19
740 Others	Inception and Adaboosted decision tree	Accuracy = 82.9%	
AUC = 0.9 (90%)	
Sensitivity = 81%	
Specificity = 84%	

Methods and Materials

Dataset description

The details of the dataset used in this article are available in Zhao et al. (2020). It consists of 347 COVID-19 images and 397 non-COVID-19 images with several types of pathology. The length of CT images ranges from 153 to 1,853 pixels with a mean of 491 pixels, whereas the width ranges from 124 to 383 pixels with a mean of 1,485 pixels. Samples of the CT images for COVID-19 and non-COVID-19 cases available in the dataset are shown in Fig. 1.

Figure 1 Samples of CT images from the dataset; (A–D) COVID-19 CT images and (E–H) non-COVID-19 CT images.

Convolutional neural networks

The CNN is a class of DL techniques that are extensively employed for solving issues regarding health informatics (Jin et al., 2020b) and in specific classification and diagnosis of medical images (Ravì et al., 2016). Thus, different CNN architectures were employed in this article. A CNN consists of several layers such as convolutional layers, pooling layers, and fully connected (fc) layers. In the convolutional layer, a number of filters of a specific size are convolved with the portion of the input image equivalent to the same size of the filter. Afterward, a feature map is generated which corresponds to the location of the features in the original image. This feature represents the spatial information of the pixel values of the input image. Next, the pooling layer down-samples the huge dimension of the feature map. Finally, the fc layer performs the classification process like the traditional artificial neural network (ANN). The key benefit of CNN over the traditional ANN, is that the feature map is now a spatial demonstration of the variables existing in the dataset, and the features produced will be significant and descriptive (Greenspan, Van Ginneken & Summers, 2016). Moreover, CNN does the feature extraction process itself, which is not the case in traditional ANN where handcrafted feature extraction techniques should be used separately then used as input to ANN (Valliani, Ranti & Oermann, 2019). Four CNNs corresponding to the state-of-the-art networks used for medical images are employed in this article. These networks include AlexNet, GoogleNet, ResNet-18, and ShuffleNet architectures. These networks have not yet been used to detect and classify COVID-19 in the literature.

AlexNet CNN

The AlexNet CNN network was introduced in 2012 by the authors in Krizhevsky, Sutskever & Hinton (2012). It was proposed after LeNet was created by LeCun et al. (1998). The authors created the AlexNet model, which won the ImageNet Large-Scale Visual Recognition Challenge in 2012. The structure of AlexNet contains 23 layers, which includes five convolutional layers, five rectified linear unit (ReLu) layers, two layers for normalization, three pooling layers, three fc layers, a layer of probabilities using softmax units, and lastly one layer for classification terminating in 1,000 neurons for 1,000 categories. AlexNet CNN was initially trained with ImageNet data, using more than 15 million labeled images with almost 22,000 classes.

GoogleNet CNN

Researchers at Google designed the GoogleNet architecture in 2014. Its construction is much deeper than AlexNet and consists of 22 layers. GoogleNet architecture is based on the inception block, which considers dropping the number of parameters in a CNN. Therefore, it contains a lower number of parameters, almost 12 times lower than AlexNet, which leads to faster convergence. The GoogleNet structure won the ImageNet Large-Scale Visual Recognition Challenge (ILSVRC14) in 2014, for achieving the best performance (Szegedy et al., 2015). The main module of this structure is nine inception blocks that occur at every layer of the GoogleNet. These blocks are scaled upon each other with a maximum pooling layer. Lastly, an fc layer exists just before the output layer.

ShuffleNet CNN

Recognizing important patterns from images, requires the construction of bigger and deeper CNN. Producing accurate CNNs regularly requires a huge number of layers and channels, which leads to a very high computation cost (Girshick et al., 2014). To overcome this problem, an extremely efficient CNN was proposed called ShuffleNet. ShuffleNet was first introduced in 2018 by Zhang et al. (2018), for applications that have very limited computing power such as mobile applications. This novel network structure presents two new processes namely; pointwise group convolution and channel shuffle. The former procedure uses 1 × 1 convolution to lower the computation cost, while obtaining acceptable accuracy compared to the state-of-the-art CNN such as ResNet and Xception, which turns out to be less effective in tremendously small networks due to the costly dense 1 × 1 convolutions. The later operation assists the data flowing across feature channels. The channel shuffle operation lets a group of convolutions attain input data from several groups, where the output/input channels are completely associated. Specifically for the feature map produced in the previous group layer, the channels in each group are split into separate subgroups. Afterward, each group of the preceding layer is fed by a separate subgroup.

ResNet-18 CNN

One of the recent architectures that are commonly used for medical imaging applications is ResNet. It received the first place in ILSVRC and the COCO 2015 competition in ImageNet Detection, ImageNet localization, Coco detection, and Coco segmentation. The main building block in ResNet is the residual block introduced by He et al. (2016). Such block offers short cut connections within the convolution layers, which allow the network to skip several convolution layers at a time. In other words, the residual block offers two choices, it can accomplish a set of functions on the input, or it can pass this phase completely. Therefore, the ResNet structure is considered more efficient and produces better performance than other CNNs such as AlexNet and GoogleNet. In this article, ResNet-18 was employed which contains 17 convolutional layers and one fully connected layer at the end of the network.

Proposed MULTI-DEEP CAD system

A novel CAD system “MULTI-DEEP”, for detecting COVID-19 and distinguishing it from non-COVID-19 is proposed, which CAD is based on the fusion of multiple CNNs. It employs several CNNs that are different from those used in the literature. It consists of four approaches; end-to-end classification, deep feature extraction, principal component selection, and detection. In the end-to-end DL classification approach, four CNNs of different architectures were constructed with the images available in the dataset. Each of the CNNs performs the classification and diagnosis procedure individually. Afterward, in the deep feature extraction approach, valuable features were extracted from each deep CNN separately. Next, in the principal component selection approach, PCA was applied on each deep feature set individually, and the optimal number of principal components was selected for each feature set separately. In the detection approach, SVM classifiers were constructed to test the performance of the proposed CAD system. The four approaches of the proposed CAD system were tested with four different scenarios, to verify that the fusion of multiple CNNs was capable of enhancing the accuracy of the CAD system and it was better than using individual CNNs. Thus, the first three scenarios were done to examine the performance of the CAD system using individual CNNs either by end-to-end DL classification or by extracting deep features from each CNN and reducing them by PCA, then using them individually to train SVM classifiers. This process was done in the first three scenarios. In the first scenario, four pre-trained end-to-end networks were fine-tuned to classify COVID-19 and non-COVID-19 cases. In the second scenario, the deep features were extracted from each CNN and then used individually to construct SVM classifiers. Additionally, in the third scenario, PCA was applied on each deep feature set separately, and then a chosen number of principal components was used to train SVM classifiers individually. The performance of the CAD system using individual CNNs (the first three scenarios) was compared with the fusion of multiple CNNs in the fourth scenario. In this last scenario, the deep features were fused to examine the influence of fusing the four deep features, and determine its impact on both detection accuracy and computational efficiency. Moreover, PCA was performed on the fusion feature set. Figure 2 illustrates the MULTI-DEEP CAD system. Note that, a pre-processing step was done to resize all the images to be equal to the required input size of the CNN architectures, as each CNN requires a different input image size.

Figure 2 The proposed different scenarios for the MULTI-DEEP CAD system.

End-to-end deep learning approach

In this approach, the transfer learning method was employed. Transfer learning is a technique that allows a pre-trained CNN that has been trained with a huge dataset containing thousands of images like ImageNet, to be used in a similar classification task. In other words, a pre-trained CNN which was previously learned with a similar task to the one at hand is used for a new medical task (Zemouri, Zerhouni & Racoceanu, 2019; Greenspan, Van Ginneken & Summers, 2016). Transfer learning is an important step to solve convergence and overfitting issues that could happen during the first few epochs. Therefore, four pre-trained CNNs of different architectures were employed. These networks include AlexNet, GoogleNet, ResNet-18, and ShuffleNet architectures. A summary of the architecture of the four networks is shown in Table 2. These CNNs were used individually to detect COVID-19 and distinguish them from non-COVID-19 cases.

Table 2 A summary of the architecture of the four pre-trained CNNs used.

CNN architecture	Number of layers	Input size	Output size	
AlexNet	8	227 × 227	4,096 × 2	
GoogleNet	22	224 × 224	1,024 × 2	
ResNet-18	18	224 × 224	512 × 2	
ShuffleNet	50	224 × 224	544 × 2	

Deep feature extraction approach

Pre-trained CNNs can either be trained from images to perform classification tasks, or be used as a feature extractor where significant deep features are extracted from the fc layers of the CNNs. In this approach, as an alternative of employing the CNNs as classifiers, deep features are pulled out from the “fc7” of the fine-tuned AlexNet, dropout layer named “pool 5 drop 7 × 7 s1” of the pre-trained GoogleNet, “node 200” of pre-trained ShuffleNet, and “global average pooling 2D layer” (fifth pooling layer) of the fine-tuned ResNet-18. The number of deep features generated from each CNN is 544, 4096, 1024, and 512 for SuffleNet, AlexNet, GoogleNet, and ResNet-18, respectively.

Principal component selection approach

The deep features extracted in the previous approach are of large dimension. Therefore, in this approach PCA is employed to reduce the dimension of the feature space vector. PCA is a popular feature reduction approach, used to compress the data size by operating a covariance investigation among variables of a dataset. It drops out the sum of observed variables to lessen the principal components. Such principle components demonstrate the variance of the observed variables (Smith, 2002). First, PCA is employed on each deep feature individually. Afterward, the optimal number of principal components for each deep feature set is chosen using a forward selection procedure. Finally, these selected numbers of principal components are fused and used to train SVM classifiers in the next approach.

Detection approach

In this approach, the detection process is performed using four different scenarios. The first scenario presents the detection of COVID-19 using the four pre-trained CNNs. The other three scenarios include the use of a cubic SVM classifier, for detecting COVID-19 and distinguishing it from non-COVID-19 cases. In the second scenario, each deep feature extracted in the deep feature extraction approach is used to construct and train four cubic SVM classifiers individually. In the third scenario, the principal components selected from the PCA applied to each deep feature set are used to train the cubic SVM classifier individually. In the fourth scenario, the effect of fusing the four deep features is investigated to determine its impact on both detection accuracy and computational efficiency. Moreover, PCA is applied to the fused features to reduce their dimension and examine if using PCA on the fused features will enhance the performance of the SVM classifier.

Experimental Set up

Parameter Setting

Several parameters are tuned for the pre-trained CNNs. The number of epochs is 20, and the initial learning rate for the four CNNs is 10−4. The mini-batch size and validation frequency are 10 and 4, respectively. The L-regularization is 0.0005. Further parameters are kept unchanged. These configurations are to confirm that the parameters are fine-tuned for the detection of medical data. Stochastic gradient descent with momentum is used for optimization. To validate the performance of the proposed MULTI-DEEP CAD system, 5-fold cross-validation is used. Note that the kernel function used for the SVM classifier is cubic, as it achieved the best performance.

Augmentation

Augmentation is an important process done to increase the size of the dataset, since training the classification model with a small amount of data might over-fit (Ravì et al., 2016). In other words, the classification of the model will memorize the details of the training set, and will not perform well on testing sets. In this article, the augmentation techniques used to create new lung CT images from the training data, are flipping, translation, and scaling. Each CT image was translated in x and y-directions with pixel range (−30 to 30). Moreover, each original image was flipped. They were also scaled with a scaling range (0.9–1.1).

Performance Evaluation

The performance of the proposed MULTI-DEEP CAD system is measured with several metrics such as accuracy, sensitivity, specificity, precision, F1-score, and area under receiving operating characteristics (AUC). The equations used to calculate such metrics are shown below Eqs. (1)–(5).

(1) Accuracy=TP+TNTN+FP+FN+TP

(2) Sensitivity=TPTP+FN

(3) Specificity=TNTN+FP

(4) Precision=TPTP+FP

(5) F1-Score=2×TP2×TP+FP+FN

where TP is the number of COVID-19 images that are correctly classified, TN is the number of non-COVID-19 images that are correctly classified. FP is the number of non-COVID-19 images that are misclassified as COVID-19. FN is the number of COVID-19 images that are misclassified as non-COVID-19.

Results

This study proposed a novel CAD system called MULTI-DEEP, based on the fusion of multiple CNNs to detect and classify COVID-19 and non-COVID-19 cases. The framework was composed of four different scenarios. In this section, each scenario is illustrated and the results of the four scenarios are discussed.

Scenario I

In this scenario, four pre-trained CNNs were used to detect COVID-19 and non-COVID-19 cases individually. The results of this scenario are shown in Table 3.

Table 3 Performance metrics of the four pre-trained CNNs of scenario I.

	Accuracy (%)	AUC	Sensitivity	Specificity	Precision	F1 score	Time	
AlexNet	73.21	0.7754	0.74	0.724	0.715	0.728	19 min 43 s	
GoogleNet	75.89	0.8166	0.762	0.756	0.753	0.758	19 min 31 s	
ResNet-18	78.29	0.8382	0.769	0.799	0.81	0.789	19 min 31 s	
ShuffleNet	71.99	0.7903	0.687	0.768	0.81	0.744	25 min 26 s	
Note:

Bold values indicate the highest results.

The highest performance was achieved by ResNet-18 with an accuracy of 78.29%, an AUC of 0.8386 (83.86%), a sensitivity of 0.769 (76.9%), a specificity of 0.799 (79.9%), a precision of 0.81 (81%), and an F1-score of 0.789 (78.9%). Whereas, the accuracies and AUC of other CNNs covered the ranges (71.99–75.89%) and (0.7754–0.8166) respectively. In addition, the sensitivity, specificity, precision, and F1-score fluctuated within the ranges (0.687–0.762), (0.724–0.768), (0.715–0.81), (0.728–0.758) respectively. Additionally, the execution time of ResNet-18 was 19 min and 31 s, which proved to be the best performance compared to other networks.

Scenario II

In this scenario, deep features were extracted from each pre-trained CNN, and were used to train SVM classifiers individually. The performance results of the SVM classifiers, constructed with deep features extracted from AlexNet, GoogleNet, ResNet-18, and ShuffleNet CNNs, are shown in Table 4.

Table 4 Performance metrics of SVM classifier trained individually with each deep feature extracted from the pre-trained CNNs.

	Accuracy (std)	AUC (std)	Sensitivity (std)	Specificity (std)	Precision (std)	F1 score (std)	Time (s) (std)	
AlexNet	90.9% (0.002)	0.95 (0)	0.922 (0.005)	0.896 (0.003)	0.891 (0.005)	0.907 (0.003)	20.991 (3.066)	
GoogleNet	89.2% (0.004)	0.95 (0)	0.914 (0.029)	0.86 (0.009)	0.849 (0.006)	0.881 (0.016)	3.867 (0.274)	
ResNet-18	92.5% (0.005)	0.97 (0)	0.933 (0.005)	0.918 (0.07)	0.916 (0.007)	0.925 (0.006)	1.947 (0.25)	
ShuffleNet	91.1% (0.002)	0.98 (0.001)	0.919 (0.003)	0.904 (0.004)	0.902 (0.005)	0.911 (0.003)	2.54 (0.168)	
Note:

Bold values indicate the highest results.

The deep features of ResNet-18 CNN achieved the highest performance compared to other CNN deep features, achieving an accuracy and AUC of 92.5% and 0.97 (97%), respectively, as shown in Fig. 3A. Moreover, the other scores attained were the highest values as well. The sensitivity, specificity, precision, F1-score, AUC, and the execution time reached 0.933 (93.3%), 0.918 (91.8%), 0.916 (91.6%), 0.925 (92.5%), and 1.947 s, respectively, as demonstrated in Table 4. Figure 3 shows the receiver operating characteristics (ROC) curve, and the computed AUC for the SVM classifier constructed with the deep features of ResNet-18 and ShuffleNet, as these networks achieved the highest results.

Figure 3 ROC curve for SVM classifier trained with deep features extracted from (A) pre-trained ResNet-18 and (B) pre-trained ShuffleNet.

Scenario III

In Scenario III, PCA was applied to each deep feature extracted from each pre-trained CNN. Then, a selected number of principal components from each deep feature set was used to train SVM classifiers individually. The number of principal components was selected in a forward sequential procedure. The accuracy of the constructed SVM classifiers with a number of principal components is shown in Fig. 4.

Figure 4 The performance of the SVM classifier constructed with various principal components.

It was noticeable from Fig. 4 that only 50 principal components for the ResNet-18 and the ShuffleNet deep features provide the highest accuracy. Therefore, 50 components were selected for deep features extracted from ResNet-18 and the ShuffleNet. However, for the GoogleNet deep features, the highest accuracy was achieved using 50 and 100 principal components. While the highest accuracy achieved for the AlexNet deep features, required 100 and 150 principal components. Therefore, four-fused feature sets were generated to train the SVM classifier and then compared to determine the best combination, which had an impact on the accuracy. Table 5 shows a comparison between the accuracies achieved by SVM classifiers, trained with the four feature sets corresponding to different combinations of selected principal components chosen from the four pre-trained CNNs.

Table 5 Performance metrics of the SVM classifiers constructed with the four feature sets formed from different combinations of the selected number of principal components.

	Accuracy (std)	AUC (std)	Sensitivity (std)	Specificity (std)	Precision (std)	F1 score (std)	Time (s) (std)	
Feature set (1) (250 features)	93.4% (0.002)	0.98 (0.001)	0.945 (0.006)	0.922 (0.003)	0.919 (0.004)	0.933 (0.003)	2.071 (0.269)	
Feature set (2) (300 features)	94% (0.002)	0.98 (0.001)	0.949 (0.001)	0.932 (0.001)	0.93 (0.001)	0.94 (0.001)	2.105 (0.001)	
Feature set (3) (350 features)	92.6% (0.003)	0.972 (0.005)	0.935 (0.006)	0.916 (0.005)	0.913 (0.005)	0.924 (0.004)	2.002 (0.194)	
Feature set (4) (300 features)	93% (0.003)	0.98 (0.001)	0.935 (0.005)	0.926 (0.005)	0.925 (0.006)	0.93 (0.001)	2.088 (0.293)	
Note:

Bold values indicate the highest results.

Feature set (1) contained 50 Components from each of ResNet-18, GoogleNet, and ShuffleNet, and 100 components from AlexNet. Whereas, Feature set (2) had 50 components from each of ResNet-18, GoogleNet, and ShuffleNet, and 150 components from AlexNet. Feature set (3) composed of 50 components from each of ResNet-18 and ShuffleNet, 100 components from GoogleNet, and 150 components from AlexNet. Feature set (4) had 50 components from each of ResNet-18 and ShuffleNet, and 100 components from GoogleNet and AlexNet. It was clear from Table 5 that the highest SVM performance was achieved when constructed using Feature set (2). Feature set (2) yields to an accuracy of 0.94 (94%), an AUC of 0.98 (98%), a sensitivity of 0.949 (94.9%), a specificity of 0.932 (93.2%), a precision of 0.93 (93%), and an F1-score of 0.94 (94%).

Scenario IV

In this scenario, the four deep features extracted from each pre-trained CNN were fused to study the influence of this fusion on the performance of the SVM classifier, and compare it with the fusion of the selected principal components of scenario III. The results of this fusion compared to scenario III were shown in Table 6.

Table 6 Performance metrics of scenario IV compared to scenario III and deep features fusion with PCA.

	Accuracy (std)	AUC (std)	Sensitivity (std)	Specificity (std)	Precision (std)	F1 score (std)	Time (s) (std)	
Fusion with PCA	92.2% (0.001)	0.97 (0)	0.93 (0.001)	0.918 (0.005)	0.916 (0.006)	0.923 (0.003)	2.368 (0.354)	
Feature set (2) (Scenario III)	94% (0.002)	0.98 (0.001)	0.949 (0.001)	0.932 (0.001)	0.93 (0.001)	0.94 (0.001)	2.105 (0.001)	
Fusion without PCA (Scenario IV)	94.7% (0.004)	0.98 (0.001)	0.956 (0.007)	0.937 (0.005)	0.934 (0.006)	0.945 (0.003)	33.765 (1.46)	
Note:

Bold values indicate the highest results.

Scenario IV results were also compared to the fusion of the four deep features after applying PCA, as shown in Table 6. The Table shows that feature set (2) of scenario III (corresponding to the fusion of 50 principal components from each ResNet-18, GoogleNet, and ShuffleNet, and 150 principal components from AlexNet), has better performance than the fusion of all deep features, after applying PCA on the fused deep features. The scores of scenario III achieved an accuracy of 94%, a sensitivity of 0.949 (94.9%), a specificity of 0.932 (93.2%), an AUC of 0.98 (98%), a precision of 0.93 (93%), and an F1-score of 0.94 (94%). However, the scores of the fusion of deep features achieved an accuracy of 92.2%, a sensitivity of 0.93 (93%), a specificity of 0.918 (91.8%), an AUC of 0.97 (97%), a precision of 0.916 (91.6%), and an F1-score of 0.923 (92.3%). Scenario IV achieved an accuracy of 94.7%, which was slightly higher than 94% of the accuracy achieved in scenario III, as shown in Table 6. This means that scenario III (corresponding to the fusion of 50 principal components from each ResNet-18, GoogleNet, and ShuffleNet, and 150 principal components from AlexNet) had almost the same performance of scenario IV, but with more efficient computational cost as the execution time of scenario III was 2.368 s, which was lower than that of scenario IV (33.765 s). A comparison between the highest achieved performances in each of the four scenarios is illustrated in Fig. 5.

Figure 5 A comparison of the highest accuracy achieved for each scenario.

Discussion

Coronavirus has led to the worst worldwide health crisis in recent history. The rapid diagnosis of COVID-19 is essential to control the spread of such a disease, and the current pandemic situation. The common method used to diagnose COVID-19 cases is the RT-PCR test, however, it has several limitations. Among these limitations are low sensitivity, insufficient availability of such a test, and the crucial need for a well-equipped laboratory for analysis (which is a key challenge especially in developing countries), delayed responses and the high cost of this test. All these limitations exposed the need for a quick alternative that enables the front-line specialists to achieve a fast and precise diagnosis.

Computed tomography imaging technique has proven to be efficient in visualizing lungs, and may facilitate the primary detection of suspected individuals. However, CT imaging alone was not enough for an accurate diagnosis. This was due to the similarity between configurations of COVID-19 and other types of pneumonia, which might confuse radiologists trying to distinguish COVID-19. Nonetheless, CAD systems based on DL techniques have shown to be more capable of discriminating between COVID-19 and non-COVID-19 cases, and can achieve an accurate diagnosis sooner than the clinical exam, which accordingly reduces the time needed for disease control (Butt et al., 2020; Shi, Han & Zheng, 2020).

The ease of accessibility of CT imagining techniques makes CT-imaging a preferable tool for coronavirus diagnosis in many counties such as China (He et al., 2020). Moreover, the cost of the diagnoses is an important factor for choosing the tool for the novel virus diagnosis. The relative low cost of CT in China makes it as well a favorable diagnostic tool (He et al., 2020). RT-PCR test is considered an expensive diagnostic tool in many countries including developing countries in Africa such as South Africa, Algeria, Egypt, Morocco, and Tunisia. This is due to the lack of availability of such test which increase its price (Kobia & Gitaka, 2020). The presently available RT-PCR kits are variable, there are more than seven platform used for the novel coronavirus diagnosis (Al-Tawfiq & Memish, 2020; Van Kasteren et al., 2020). Such tests may result in initial false negative outcomes (Long et al., 2020). Examples of such kits are Altona Diagnostics, BGI, CerTest Biotec, KH Medical, Primer Design, R-Biopharm AG, and Seegene (Van Kasteren et al., 2020). RT-PCR platforms are presenting sensitivities ranging between 45% and 60%; therefore, specifically in the initial course of an infection, the test repetition may be essential to achieve a diagnosis. This is difficult to attain with the global shortage of testing kits specially in developing counties (Al-Tawfiq & Memish, 2020). Moreover, RT-PCR tests may take few hours to complete, however they are restricted by the time needed to transfer and arrange sample for testing. Nowadays, due to the increase in the demand of such test, the laboratories have been flooded with samples leading to significant delays in diagnostic duration (Kim, Hong & Yoon, 2020; Ai et al., 2020; Young et al., 2020). In addition, the availabilities of such labs are limited in some developing countries like Egypt, in contrast to the CT machines, which are located in many hospitals and scan centers. The authors in Shen et al. (2020) showed that CT results are obtainable in time which are lower than RT-PCR test. Also, Ai et al. (2020) compared the speed accuracy, and sensitivity of TaqMan One-Step RT-PCR kits with CT and the results showed that CT is more sensitive and faster than RT-PCR kit used. This make the use CT more preferable specially when used with artificial intelligence techniques such DL (Kobia & Gitaka, 2020). Several studies showed that using CAD systems based on DL and CT images are considered a rapid, more accurate, and sensitive than the RT-PCR test. This is because when a CT image enters the CAD system, it is automatically classified to COVID-19 and non-COVID-19 in less than an hour with a better accuracy and sensitivity than the RT-PCR test (Das et al., 2020; Manigandan et al., 2020; Song et al., 2020; Maghdid et al., 2020).

In this article, we proposed a fast solution, that may serve as an alternative to the RT-PCR test. The proposed solution introduced a novel CAD system, based on the fusion of multiple CNNs trained with CT images. This study is a crucial trial comprising a simple construction, low cost, and automatic CAD system that can achieve a high accuracy, based on the fusion of multiple CNNs. During these difficult days of the worldwide pandemic, the proposed MULTI-DEEP CAD system has great potential to be used as a tool for COVID-19 detection. It was verified that CT provides a convenient and effective method to primary identify suspicious cases (Xu et al., 2020; Long et al., 2020). Thus, the proposed system can detect COVID-19 cases early, thereby avoiding the fast spread of the disease. The proposed CAD system is a CNN-based model. CNN methods provided that the characteristic lung lesions exist at the time of testing are more capable of distinguishing between COVID-19 and non-COVID-19 cases than a manual diagnosis from CT images (Butt et al., 2020).

The outperformance of CNN was proven in the related studies by various authors (Bai et al., 2020; Chen et al., 2020; Ardakani et al., 2020), who compared the performance of their CAD system based on DL techniques, with that of a trained radiologist without the help of a CAD system. These studies indicated that the performance of DL based CAD systems outperformed manual diagnosis by a trained radiologist without the help of a CAD system. The authors in Bai et al. (2020) showed that their DL based CAD system achieved higher test accuracy (96% vs. 85%), sensitivity (95% vs. 79%), and specificity (96% vs. 88%) than radiologists. Whereas in Ardakani et al. (2020) the performance of the radiologist was lower than the authors’ proposed DL-based CAD system, with a sensitivity of (89.21% vs. 98.04%), specificity of (83.33% vs. 98.04%), and accuracy of (86.27% vs. 99.5%). On the other hand, in Chen et al. (2020), the authors showed that their DL based CAD system can reduce the manual radiologist’s diagnosis time by 65%.

As it was obvious from Fig. 5 that scenario III, corresponding to the fusion of the selected number of principal components from each deep feature set, has increased the accuracy to 94% compared to 92.5% achieved in scenario II using SVM classifier trained using only deep features extracted from ResNet-18 CNN. Although, the accuracy was slightly improved to 94.7% when fusing all deep features extracted from the four pre-trained CNNs without PCA, the execution time in scenario III is 2.105 s, which was lower than the 33.765 s of scenario IV, and much lower than the 19 min 31 s of scenario I. This means that the fusion of the selected principal components led to a more efficient CAD system.

MULTI-DEEP has achieved accuracy, AUC, sensitivity, and specificity of 94.7%, 98%, 95.6%, and 93.7% for identifying COVID-19 and non-COVID-19 cases. By comparing the results of the proposed MULTI-DEEP CAD system with the related work presented in Table 1, it was noticed that the CNN architectures used were different from those used in MULTI-DEEP. Regarding the ResNet CNN architectures, it was obvious that the researchers used it with different layer structures than in Butt et al. (2020), Li et al. (2020a), Jin et al. (2020a, 2020b), and Song et al. (2020). Butt et al. (2020) used the ResNet-18 and ResNet-23 CNNs to classify COVID-19 and non-COVID-19 samples, achieving an accuracy of 86.7%, which is lower than that of MULTI-DEEP. On the other hand, Jin et al. (2020a) performed the classification using ResNet-152 CNN reaching an accuracy of 94.8%, which is almost the same accuracy of MULTI-DEEP, which was constructed with a fewer number of images. It is expected that the performance of the proposed MULTIDEEP will be enhanced when the number of images is increased. Shi, Han & Zheng (2020), Song et al. (2020), and Jin et al. (2020b) used the ResNet-50 CNN achieving an accuracy of 89.5%, 86%, and 94.8%, respectively. The accuracies achieved by Li et al. (2020a) and Song et al. (2020), are much lower than the proposed MULTI-DEEP, but the accuracy attained by Jin et al. (2020b) is almost the same as the proposed method and this is because of the large amount of data employed by Jin et al. (2020b). Furthermore, when Amyar, Modzelewski & Ruan (2020) and Chen et al. (2020) employed the U-Net for the segmentation and/or classification processes, the accuracy reached 86%, and 95.2% respectively. The high accuracy achieved by Chen et al. (2020) is due to the huge number of images used to train their U-NETs compared to the proposed MULTI-DEEP and this is one of the limitations of MULTI-DEEP. However, MULTI-DEEP achieved acceptable performance, even with a smaller number of images. Zheng et al. (2020) employed a U-Net for segmenting CT images, and then the authors constructed a CNN with eight layers. The accuracy attained was 90.9%, which was much lower than achieved in MULTI-DEEP.

It is quite obvious from the previous comparison that the proposed MULTI-DEEP performs competitively, compared to the most recent CAD systems shown in the literature. The results of MULTI-DEEP verify that it can be used to overcome the limitation of using CT imaging alone to diagnose the COVID-19 virus. The satisfactory accuracy achieved by the proposed model, even with its primary outcomes, unlocks the door for the production of a comprehensive product by IT (information technology) solution companies. This product can work mobile and appeal to the end-user, which are the radiologists or medical experts, to facilitate the diagnosis process of the COVID-19. It can be made as a mobile application or webpage, where a CT image is imported to the application or webpage and a decision is exported.

The possible scenario where this algorithm can be used, is when a patient is suffering from any coronavirus symptoms. The medical expert can take a CT scan of them instead of the RT-PCR test, then import the CT image into the proposed CAD system for a quicker, more accurate, simpler, and less expensive decision. This decision made by MULTI-DEEP will assist the specialist in improving the quality of patient management, even in extraordinary workload situations. As mentioned before, the proposed CAD system is preferable to the RT-PCR test, due to the various limitations of the latter test. The proposed CAD system can be tested on any suspected patient to determine if they are infected with COVID-19.

A limitation of this study that could be considered in future research, is the relatively small number of COVID-19 images used. The performance of the proposed CAD system is likely to improve when the training data is increased. Besides, at the moment the proposed model can only distinguish COVID-19 from non-COVID-19 cases. It is essential to distinguish COVID-19 cases from other types of pneumonia as well. Future research will be done to expand the model so that it can correctly diagnose other types of pneumonia. One more limitation is that the performance of the CAD system was not compared with a trained radiologist. Future work will also focus on using multimodal imaging techniques, using more types of CNNs, and finally employing a segmentation technique to distinguish lung from other tissue, and comparing its performance to that of the proposed CAD system.

Conclusions

Several studies have verified that DL can help radiologists to accurately diagnose COVID-19. The key goal of this article was to construct an efficient CAD system capable of accurately detecting COVID-19, and differentiating it from non-COVID-19 cases. The proposed MULTI-DEEP CAD system was based on the fusion of multiple CNNs. It went through four experiments to achieve a more efficient CAD system. The experiments showed that fusing deep features extracted from several pre-trained CNNs, improved the performance of the proposed CAD system. The experiments also proved that fusing the optimal number of principal components selected from each deep feature, has decreased the computational cost of the CAD to almost 32%, which lead to a more efficient CAD system. The results verified that the proposed MULTI-DEEP CAD system could successfully detect COVID-19 with high accuracy. Additionally, it showed its competitive performance compared to similar studies. Therefore, MULTI-DEEP can be employed by radiologists to facilitate the diagnosis process of COVID-19. MULTI-DEEP may assist in managing the current pandemic, speed up the detection of the disease, avoid its fast spread, and help doctors to improve the quality of patient management, even in extraordinary workload situations. MULTI-DEEP should undergo further field testing before it can be employed by radiologists. In addition, it will likely have to undergo regulatory approval by health authorities before its implementation into hospitals.

Supplemental Information

Supplemental Information 1 The matlab code.

Click here for additional data file.

Additional Information and Declarations

Competing Interests

Author Contributions

Data Availability

The authors declare that they have no competing interests.

Omneya Attallah conceived and designed the experiments, analyzed the data, prepared figures and/or tables, authored or reviewed drafts of the paper, and approved the final draft.

Dina A. Ragab conceived and designed the experiments, performed the experiments, analyzed the data, prepared figures and/or tables, authored or reviewed drafts of the paper, and approved the final draft.

Maha Sharkas conceived and designed the experiments, authored or reviewed drafts of the paper, and approved the final draft.

The following information was supplied regarding data availability:

Data is available at GitHub (COVID-CT-Dataset): https://github.com/UCSD-AI4H/COVID-CT.

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
