# Peer review of "MULTI-DEEP: A novel CAD system for coronavirus (COVID-19) diagnosis from CT images using multiple convolution neural networks"

_PeerJ, doi:10.7717/peerj.10086_

## Round 0.1 · original submission · Major Revisions

Study design, diagnostic test evaluation and machine learning experts have reviewed your manuscript and identified several issues that require your attention before it can be considered for publication. In addition, details including your code are missing which prevents reviewers from verifying your results, please include in the next revision your code and the data required to run this code.

Reviewer 1 ·

Basic reporting

Basic reporting
Introduction
The introduction should emphasize more clearly what the advantage of a rapid diagnosis of Covid19 is. The advantages mentioned are a bit vague (to lessen the load on the healthcare organization as well as achieving the finest potential care for infected people). As far as I understand, there is only supportive care available for Covid19 patients apart from experimental treatments, but I’m sure this situation is fluid. Is the advantage to the swab test the speed of getting results? Does this help with isolation protocols and saving resources in case of negative results? Would someone testing negative on the CAD not be put in isolation, knowing that the test does not have 100% sensitivity? Are patients diagnosed based on CT scans alone? Or are CT scans mostly used to assess severity? Do patients with mild symptoms have lung abnormalities? I’m just giving examples of questions, they need not necessarily be addressed, but as a reader I’m not sure I understand what the advantage and relevance of CAD for Covid19 is.
The introduction lists a number of previously developed machine-learning models to diagnose Covid19 from CT scans. There is mention of multiple methods called ResNet50, U-Net++ and others. For a reader in the medical field who is not familiar with the jargon these names mean nothing. Can you give a short explanation of what these are?
I recommend moving the paragraph 2.2 in Materials and Methods to the introduction as a description of what CNNs are.
The language is often awkward. I recommend having the manuscript edited by a native English speaker. Here are some examples:
• Line 29 COVID-19 is generated initially in Wuhan, China – this sentence should be rephrased to something like: COVID19 was first observed in Wuhan, China
• Line 120: … “is not investigated”. Do you mean the effect was not investigated in those previous studies?
• Line 121: This is an incomplete sentence
• Lines 144/145: is it convolutional layer or convolution layer?
Specific items:
Line 57: Is there evidence for this statement? If yes, please insert reference; if not, please state that this is a speculation
Line 119: Please define DCNN

Experimental design

Experimental design
Materials and Methods
How was it determined that the images are or are not from Covid19 patients? What was the reference test? I briefly looked at the reference, which is a webpage with a link to the dataset. I could not determine what methods were used to distinguish cases from non-cases. If you have the information, please state it briefly; if you don’t have it then also please comment on the lack of that information.
Furthermore, can you describe the data set of images in terms of severity, e.g. hospitalized patients, a range of severities? Again, if that info is not available, please state so.
Other authors have applied segmentation to distinguish lung from other tissues. Has this been done in your method as well? If yes, where is it described? If no, why not?
Line 264: You mention that you are using 5-fold cross validation. Are the reported performance metrics based on testing with part of the data set aside for this purpose and not used during model training/validation? Or is the entire data set used for training/validation/testing?
Specific issues:
Line 205: here you say the proposed CAD is based on the fusion of multiple CNNs. Then in line 296 you say four pre-trained CNNs are used to detect COVID-19 and non-COVID19 cases individually. Are the CNNs fused or working individually? It seems from the results that they are working individually, so the statement in line 205 needs clarification.
Line 134: is the length and width of CT images unitless?
Line 145: is FC layers = fully connected? Please define at first mention
Figure 2:
What is the pre-processing step in this figure? I don’t see this mentioned in the text.
The resolution of the image should be improved. After reading the paper several times, I’m still a bit confused about your use of the terms stage and scenario. Is the process described a sequence of processes from I to IV or are these different scenarios, i.e. one of the options is chosen?
The diagram looks as if the 4 scenarios are either happening simultaneously or one is picked by the system. In the description of the text, it sounds like all these steps are happening sequentially. Which one is it? The word scenario may not be very good if all steps happen each time; the word stage may be better in this context. Or are there 4 stages and in stage 4 there are 4 different scenarios as mentioned in Line 247 where the scenarios represent increasing numbers of combination of stages?
Table 2: what are the units of the input and output sizes?

Validity of the findings

Results
The first paragraph seems to be a summary of the methods – consider eliminating.
Table 3 and 4: it looks like you have (std) in both table headings but only results in Table 4. Also, time (sec) seems not appropriate in the Table 3 heading.
Discussion
You are mostly summarizing the results in your discussion without critically comparing your work to that of others. Some of the questions your discussion should attempt to answer are:
• What are the limitations of your study?
• What is a possible scenario where this algorithm can be used and how will it be implemented? Obviously all patients that get a CT scan for diagnosis of Covid19, but who are those patients? What distinguishes them from those that get a swab test?
• What are the next steps?
• The uses you are stating are a bit lofty, i.e. decrease death rates, reduce load on the healthcare organization and improve quality of health management. What is your estimate of the impact this system might achieve? How many more patients will be accurately diagnosed based on the CAD than if this was not available?
• Do you know what the performance of a trained radiologist is without the help of a CAD (Se, Sp, etc)?
• Who implements these systems into practice? How do these CAD systems become incorporated into CT software?

Additional comments

The topic of the study is very timely and important and has the potential to affect a large number of patients worldwide. A novel machine learning algorithm is described for the diagnosis of Covid19 pneumonia based on a CT scan that could potentially make important contributions to this field. Your results look promising, but the process needs to be described more clearly. In particular, the four stages or scenarios are quite confusing. Figure 2 should be produced with higher resolution, at the moment it is hard to read. This figure should also be used to more clearly describe how the system works, i.e. what is a stage, what is a scenario, how does the information flow?

·

Basic reporting

There are a number of concerns about the manuscript, including the following:#

1) The manuscript has some merits, but there are too many grammatical and typo errors throughout, but especially in the Introduction. This detracts from the paper. Some examples are given below. This should be fully addressed.

2) Line 1: Title of paper – this should be “… diagnosis of …”.
3). Line 29 Abstract. Change ‘generated’ to ‘originated’.
4). Line 50-53. Include the AUC value for your method as part of the metrics.
5). There are a number of factual inaccuracies and speculation. E.g. in Line 57. The authors are speculating about the origin of SAR-COV2 (i.e. from bat). This is inappropriate as this has not been conclusively proven.
6) Line 61: it is incorrect to say that WHO declared COVID-19 a pandemic on 11 March. It was actually at the end of January 2020. Note also that the correct phrase is that it was declared a pandemic. It was stated as a pandemic is incorrect phrase.
7). Line 129: This is an incomplete statement. If you mean to say that Table 1 should be inserted here, then say so, otherwise delete line. E.g. (Please insert Table 1 here).
8). Line 138-139. Similar comments to Line 129 above.

9) Line 175: AlexNet is one of the primaries of CNN. This does not make sense. Unclear what is meant by this.
10) Line 216 See comments to Line 129 above.

11). Line 203-215 Section 2.3 You should give a good overview of your system here. Explain clearly the role of each block and how the overall system is used to achieve the detection of COVID-19.

12). Line 304 See previous comments.
13). Line 315-318. These are where the figures are to be inserted. If so make this clear (e.g. Please insert table 4 here)
14) Line 366- Section 6 Discussions. The English in this section need attention.
15) Line 394. See previous comments on insertion of figures and tables.

20. Line 315-318. These are where the figures are to be inserted. If so make this clear (e.g. Please insert table 4 here)

Experimental design

1).. The authors have given a good review of the literature in the Introduction. However, in Line 120-121 the authors appear to be saying that the main issue with existing methods is that the effects of combining multiple CNNs on classification/diagnosis has not been investigated, together with the need to reduce the feature space. Given that these are the main contributions and novelties in the paper, the authors should expand these points in both the Introduction and in the Discussions.

2).. The CT shown for COVID-19 and non-COVID-19 are quite distinct. Why do we need the complicated structure of proposed in your methods to distinguish between them?

3). Section 2.31 Why is it appropriate to use these particular pre-trained models, given that the classification task in COVID-19 and in the original task for these CNNs are different?

4). Line 266-273 Section 3.2 Augmentation. What are the final sizes for the datasets (COVID-19 and non-COVID019 sets) after augmentation and what is the rationale for settling for these.

Validity of the findings

1). In Table 1, one of the best models has performance metrics of AUC = 0.996, Sensitivity of 98.2% and specificity of 92.2%. Some of the existing methods appear to perform better than your method in some of the metrics. A comment on this should be included in the Discussions and Abstract.

2). Your method uses publicly available CNN models, including those developed by commercial companies. You should discuss the implications of this in clinical practice.

3) Line 341. Please give the AUC to allow effective comparison

4). Line 403-407. According to Table 1, it appears that some of the existing methods have better or comparable performance metrics than MULTI-DEEP. What are then the benefits of MULTI-DEEP?

5). There should be comments on the novelties/contributions of the study in relation to existing methods in the Discussions.

6). It is unclear how the outcome of the study can be translated into clinical practice, given that some of the CNN tools are owned by a commercial company and the issue of confidentiality of patient information. This point should be discussed.

Additional comments

The software codes for the study should be made publicly available.

---

## Round 0.2 · Minor Revisions

Please address the reviewers comments, I am in agreement that the suggested changes will improve the manuscript clarity and get it closer to a final version.

Reviewer 1 ·

Basic reporting

There are still some issues with English in the manuscript. I've pointed out a few instances, but there are a number of awkward phrases or grammatical errors.

Line 52: I recommend changing the wording to “the WHO encouraged efforts to reduce the spread of the infection, many countries…”
Line 55: “led to a tremendous increase in the need for hospital beds and necessary medical equipment”
Line 56:” Moreover, doctors and nurses are at high risk of infection.”
Line 63: “ the sensitivity of the RT-PCR test is inadequate resulting in more infections through test negative patients.”
Line 67: “precisely at the center of the pandemic area” – what do you mean with this phrase? The center of the pandemic is hard to pinpoint and is all around the world but also constantly shifting.
Line 72: use “advantage” instead of “privilege”.
Line 73: “deducing a more favorable illustration” – I don’t understand this phrase
Line 80: “help radiologists in giving an accurate diagnosis based on medical images”
Line 96: “DL methods are more appropriate for detecting covid-19 from CT images” – do you mean more appropriate than a radiologist? Or should they be used to enhance the radiologist’s diagnostic capabilities?
Line 155, , 174: convolution layer – you said in your rebuttal it was convolutional layer – please correct.
Line 168: “In the literature, these networks have not yet been used to detect and classify COVID-19”
Line 177: “AlexNet CNN was initially trained with ImageNet data…”
Line 197: “to lower the computation cost”
Line 402 “Discussion” instead of “Discussions”
Line 403 Delete the word “Nowadays” – it means “at the present time, in contrast with the past” Suggest changing the wording to “COVID-19 has led to the greatest worldwide health crisis in the recent past.”
Line 438 – 450 This section is a repeat of your methods and has already been explained at length before. Please delete. Figure 5 is part of the results and should be described in the results section.
Line 458 Your missing a digit, should be 33.765 s
Line 505 I wouldn’t use “inadequate” – use “relatively small”. Inadequate implies that you really didn’t have enough images to do the work

Experimental design

Your narrative of the design would be a lot easier to read if you presented it in the past tense as a description on how you developed and tested the system. The use of the present tense throughout your description implied to me that the entire process is part of MULTI-DEEP, when in reality, as I understand, it is scenario 3, which maximizes accuracy and computational speed. Please describe the development of the system in past tense instead of describing what you did in the present tense. E.g., line 221: “In the end-to-end DL classification stage, four CNNs of different architectures were constructed with the images available in the dataset. Each of the CNNs performed the classification and diagnosis procedure individually. In deep feature extraction, valuable features were extracted from each CNN separately…”
I still found the term “stage” confusing in your narrative, as it implies that subsequent stages depend on previous stages. I would prefer a term like “component” or “approach” to enhance clarity.

Validity of the findings

Line 507: I think you mean that you want to expand the model so that it can correctly diagnose other types of pneumonia while at the moment it can only distinguish COVID-19 from non COVID-19 cases. If so, please change the wording accordingly.

Conclusions

You should add that MULTI-DEEP should undergo further field testing before it can be employed by radiologists. Also, it will likely have to undergo regulatory approval by health authorities before its implementation into hospitals.

Additional comments

Thank you for putting a lot of effort into the revision of the manuscript. I have a few more comments at this point.

·

Basic reporting

The English has improved, but there is still a need to still proof read the revised manuscript.

Experimental design

This is satisfactory.

Validity of the findings

This is satisfactory.

Additional comments

The authors have addressed most of my concerns.
However, it is recommended that a further proof reading of the manuscript is undertaken.

---

## Round 0.3 · Minor Revisions

Thank you for addressing the reviewer comments, I agree that the manuscript's English language has improved greatly. I do however have the following comments that should be addressed before it is suitable for acceptance:

1- While you identify the source of the data, your Matlab file is missing the data input command(s), otherwise its not clear if the .m file uptakes the images directly or some pre-processing is needed before it can be run. Please update the .m file with all the commands needed once Zhao et al's image repository is acquired.

2- Because the proposed DL algorithms were not compared head to head to any COVID-19 rt-PCR, the manuscript lacks the data for the statement in line 420 of the untracked manuscript version "In this paper, we proposed a faster solution, as an alternative to the RT-PCR test. ". Not to be pedantic, but would this apply to all COVID-19 PCR tests, there are numerous ones used, the WHO, US, Italian, amongst many others. Also at what level of disease progression in patients does this statement stand true, would it be still true if we are comparing early clinical cases with less advanced lung lesions?

This statement should be edited to something along the lines of:
"In this paper, we proposed a fast solution, that may serve as an alternative to RT-PCR tests." (notice fast, instead of faster since you didn't present data comparing time from test to results for both, also notice may be an alternative instead of "an alternative") Similarly:
- line 785 of the tracked manuscript, reword CAD may be preferable instead of "is".
- line 787 the sentence should be modified to indicate that it would be true provided the characteristic lung lesions exist at the time of testing.

3- The discussion section should also include a serious discussion of the differences between the two test modalities, cost of CT-Scan which can be more than PCR in a lot of countries, and at times, depending on the platform may not be any quicker than PCR. (just as an example, Fluidigm's Biomark platform running the CDC's PCR can now test 192 patient specimens for up to 24 targets (so not only detect COVID, but also differentiate between multiple viruses including COVID-19, Flu, H1N1 and several others) all within 2-4 hours and for a cost that is less than a single CT-scan.

4-In line 349 of the tracked manuscript how did you identify the optimal number of components, what loading cut-off was used for example? This info should be added to the manuscript as its not clear if it is a manual step that requires the investigator's input or automated.

5-A few edits are needed:
-line 94 of tracked manuscript, deleted the "a"
-line 109 of tracked manuscript, replace done through with done during. Also what kind of examination, I am assuming clinical? And visual fatigue here is referring to the radiologist? If so reword for clarity please.
-line 170-1 of tracked manuscript its not clear if these estimates are for Unets.
- line 630 of tracked manuscript, I think by pathogenic exam you mean clinical exam, if so reword to latter.

---

## Author Rebuttal · Round 0.3

Arab Academy for Science, Technology, and Maritime Transport (AASTMT)
College of Engineering and Technology
Alexandria, 1029, Egypt.

August 24th, 2020

Dear Editors

We would like to thank the editor and the reviewers for their generous comments on the manuscript that have helped us in improving the quality of this manuscript. In addition, we have edited the manuscript to address their concerns.
We believe that the manuscript is now suitable for publication in PeerJ.

Yours sincerely;

Omneya Attallah and Dina A. Ragab

## Editor comments (Sharif Aly)

*Please address the reviewers comments, I am in agreement that the suggested changes will improve the manuscript clarity and get it closer to a final version.*

Thank you for encouraging us to update our paper and to submit the revised version. We find that all the comments are addressable and these comments have helped us to improve the quality of our work.

# Reviewer 1

## Basic reporting

*There are still some issues with English in the manuscript. I've pointed out a few instances, but there are a number of awkward phrases or grammatical errors.*

*Line 52: I recommend changing the wording to "the WHO encouraged efforts to reduce the spread of the infection, many countries…"*
*Line 55: "led to a tremendous increase in the need for hospital beds and necessary medical equipment"*
*Line 56:" Moreover, doctors and nurses are at high risk of infection."*
*Line 63: " the sensitivity of the RT-PCR test is inadequate resulting in more infections through test negative patients."*
*Line 72: use "advantage" instead of "privilege".*
*Line 80: "help radiologists in giving an accurate diagnosis based on medical images"*
*Line 168: "In the literature, these networks have not yet been used to detect and classify COVID-19"*
*Line 177: "AlexNet CNN was initially trained with ImageNet data…"*
*Line 197: "to lower the computation cost"*

**Response:** Thank you for the valuable corrections. We updated the manuscript by addressing all these comments. See lines 55-56, 58-60, 66-67, 76, 86, 186, 196, and 218.

*Line 155 , 174: convolution layer – you said in your rebuttal it was convolutional layer – please correct.*
*Line 402 "Discussion" instead of "Discussions"*
*Line 458 Your missing a digit, should be 33.765 s*
*Line 505 I wouldn't use "inadequate" – use "relatively small". Inadequate implies that you really didn't have enough images to do the work*

**Response**: Corrected with thanks.

*Line 403 Delete the word "Nowadays" – it means "at the present time, in contrast with the past"*
*Suggest changing the wording to "COVID-19 has led to the greatest worldwide health crisis in the recent past."*

**Response**: Corrected with thanks.

*Line 438 – 450 This section is a repeat of your methods and has already been explained at length before. Please delete. Figure 5 is part of the results and should be described in the results section.*

**Response:** This section is deleted, and Figure 5 is described in the results section.

*Line 67: "precisely at the center of the pandemic area" – what do you mean with this phrase? The center of the pandemic is hard to pinpoint and is all around the world but also constantly shifting.*

**Response:** We removed this phrase "precisely at the center of the pandemic area." See line 78.

*Line 73: "deducing a more favorable illustration" – I don't understand this phrase*

**Response:** We replaced it to "DL methods have the ability to present optimum representations and significant information from the raw images without image enhancement, segmentation, and feature extraction processes, which correspondingly offers improved diagnosis process and lower complexity than classical ML approaches." See lines 77-81.

*Line 96: "DL methods are more appropriate for detecting covid-19 from CT images" – do you mean more appropriate than a radiologist? Or should they be used to enhance the radiologist's diagnostic capabilities?*

**Response:** We paraphrased this sentence to "DL methods could be used to enhance the radiologist's diagnostic capabilities of COVID-19 from CT images." See lines 104-105.

## Experimental design

*Your narrative of the design would be a lot easier to read if you presented it in the past tense as a description on how you developed and tested the system. The use of the present tense throughout your description implied to me that the entire process is part of MULTI-DEEP, when in reality, as I understand, it is scenario 3, which maximizes accuracy and computational speed. Please describe the development of the system in past tense instead of describing what you did in the present tense. E.g., line 221: "In the end-to-end DL classification stage, four CNNs of different architectures were constructed with the images available in the dataset. Each of the CNNs performed the classification and diagnosis procedure individually. In deep feature extraction, valuable features were extracted from each CNN separately…"*

**Response:** Thank you for the comment. We updated the manuscript and we rewrote the explanation of the system in past tense.

*I still found the term "stage" confusing in your narrative, as it implies that subsequent stages depend on previous stages. I would prefer a term like "component" or "approach" to enhance clarity.*

**Response:** We replaced the term "stage" to "approach" through the whole manuscript.

## Validity of the findings

*Line 507: I think you mean that you want to expand the model so that it can correctly diagnose other types of pneumonia while at the moment it can only distinguish COVID-19 from non COVID-19 cases. If so, please change the wording accordingly.*

**Response:** We paraphrased this phrase to "at the moment the proposed model can only distinguish COVID-19 from non COVID-19 cases. It is essential to distinguish COVID-19 cases from other

types of pneumonia as well. Future research will be done to expand the model so that it can correctly diagnose other types of pneumonia." See line 551-555.

## Conclusions

*You should add that MULTI-DEEP should undergo further field testing before it can be employed by radiologists. Also, it will likely have to undergo regulatory approval by health authorities before its implementation into hospitals.*

**Response:** Thank you for the comment. We added this paragraph in the conclusion.

## Comments for the author

*Thank you for putting a lot of effort into the revision of the manuscript. I have a few more comments at this point.*

**Response:** Thank you so much.

# Reviewer 2 (Emmanuel Ifeachor)

## Basic reporting

*The English has improved, but there is still a need to still proof read the revised manuscript.*

Thank you very much for the comment. A native English speaker has revised and edited the revised version of this manuscript and we are now confident about the language.

## Experimental design

*This is satisfactory.*

**Response:** Thank you so much.

## Validity of the findings

*This is satisfactory.*

**Response:** Thank you so much.

## Comments for the author

*The authors have addressed most of my concerns. However, it is recommended that a further proof reading of the manuscript is undertaken.*

**Response:** Thank you so much. A native English speaker has revised and edited the revised version of this manuscript and we are now confident about the language.

---

## Round 0.4 · accepted · Accept

Thank you for addressing the reviewers' and my comments, I am glad to inform you that your manuscript has been accepted for publication, Congratulations.
Best wishes,
Sharif Aly